# Broad-Coverage Transition-Based UCCA Parsing

## Abstract

We present the first parser for UCCA, a cross-linguistically applicable framework for semantic representation, which builds on extensive typological work, and supports rapid annotation. UCCA poses a challenge for existing parsing techniques, as it exhibits reentrancy (resulting in DAG structures), discontinuous structures and non-terminal nodes corresponding to complex semantic units. To our knowledge, the conjunction of these formal properties is not supported by any existing parser. Our transition-based parser, which uses a novel transition set and features based on bidirectional LSTMs, has value not just for UCCA parsing: its ability to handle more general graph structures will inform the development of parsers for other semantic DAG structures, and in languages that frequently use discontinuous structures.

## 1 Introduction

Universal Conceptual Cognitive Annotation (UCCA, Abend and Rappoport, 2013) is a cross-linguistically applicable semantic representation scheme, building on the established Basic Linguistic Theory typological framework (Dixon, 2010a,b, 2012), and Cognitive Linguistics literature (Croft and Cruse, 2004). It has demonstrated applicability to multiple languages, including English, French, German and Czech, support for rapid annotation, and stability under translation (Sulem et al., 2015). It has also proven useful for machine translation evaluation (Birch et al., 2016). UCCA differs from syntactic schemes in terms of content and formal structure. It exhibits reentrancy, discontinuous nodes and non-terminals, which no single existing parser

supports. Lacking a parser, UCCA's applicability has been so far limited, a gap this work addresses.

We present the first UCCA parser, TUPA (Transition-based UCCA Parser), building on recent advances in discontinuous constituency and dependency graph parsing, and further introducing novel transitions and features for UCCA. Transition-based techniques are a natural starting point for UCCA parsing, given the conceptual similarity of UCCA's distinctions, centered around predicate-argument structures, to distinctions expressed by dependency schemes, and the achievements of transition-based methods in dependency parsing (Dyer et al., 2015; Andor et al., 2016; Kiperwasser and Goldberg, 2016). We are further motivated by the strength of transition-based methods in related tasks, including dependency graph parsing (Sagae and Tsujii, 2008; Ribeyre et al., 2014; Tokgöz and Eryiğit, 2015), constituency parsing (Sagae and Lavie, 2005; Zhang and Clark, 2009; Zhu et al., 2013; Maier, 2015; Maier and Lichte, 2016), AMR parsing (Wang et al., 2015a,b, 2016; Misra and Artzi, 2016; Goodman et al., 2016; Zhou et al., 2016; Damonte et al., 2017) and CCG parsing (Zhang and Clark, 2011; Ambati et al., 2015, 2016).

We evaluate TUPA on the English UCCA corpora, including in-domain and out-of-domain settings. To assess the ability of existing parsers to tackle the task, we develop a conversion procedure from UCCA to bilexical graphs and trees. Results show superior performance for TUPA, demonstrating the effectiveness of the presented approach.[1]

The rest of the paper is structured as follows: Section 2 describes UCCA in more detail. Section 3 introduces TUPA. Section 4 discusses the data and experimental setup. Section 5 presents

---

[1] Our code and models will be made freely available upon publication.

the experimental results. Section 6 summarizes related work, and Section 7 concludes the paper.

## 2 The UCCA Scheme

UCCA graphs are labeled, directed acyclic graphs (DAGs), whose leaves correspond to the tokens of the text. A node (or *unit*) corresponds to a terminal or to several sub-units (not necessarily contiguous) viewed as a single entity according to semantic or cognitive considerations. Edges bear a category, indicating the role of the sub-unit in the parent relation. Figure 1 presents a few examples.

UCCA is a multi-layered representation, where each layer corresponds to a "module" of semantic distinctions. UCCA's *foundational layer*, targeted in this paper, covers the predicate-argument structure evoked by predicates of all grammatical categories (verbal, nominal, adjectival and others), the inter-relations between them, and other major linguistic phenomena such as coordination and multi-word expressions. The layer's basic notion is the *scene*, describing a movement, action or state. Each scene contains one main relation (marked as either a Process or a State), as well as one or more Participants. For example, the sentence "After graduation, John moved to Paris" (Figure 1a) contains two scenes, whose main relations are "graduation" and "moved". "John" is a Participant in both scenes, while "Paris" only in the latter. Further categories account for inter-scene relations and the internal structure of complex arguments and relations (e.g. coordination, multi-word expressions and modification).

One incoming edge for each non-root node is marked as *primary*, and the rest (mostly used for implicit relations and arguments) as *remote* edges, a distinction made by the annotator. The primary edges thus form a tree structure, whereas the remote edges enable reentrancy, forming a DAG.

While parsing technology in general, and transition-based parsing in particular, is well-established for syntactic parsing, UCCA has several distinct properties that distinguish it from syntactic representations, mostly UCCA's tendency to abstract away from syntactic detail that do not affect argument structure. For instance, consider the following examples where the concept of a scene has a different rationale from the syntactic concept of a clause. First, non-verbal predicates in UCCA are represented like verbal ones, such as when they appear in copula clauses or noun phrases. Indeed,

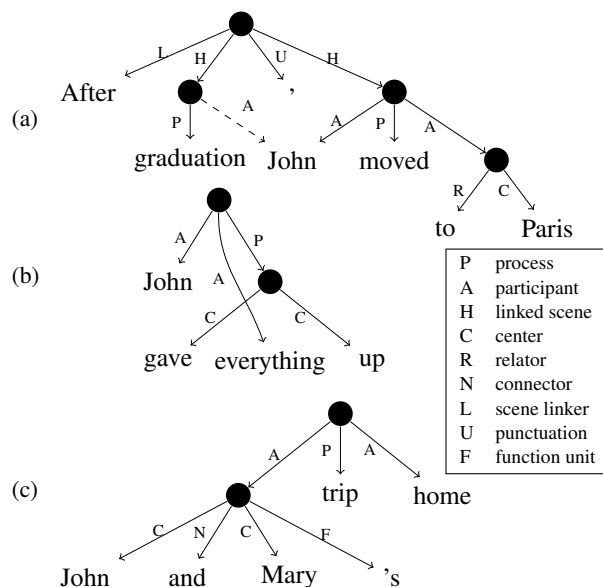

Figure 1: UCCA structures demonstrating three structural properties exhibited by the scheme. (a) includes a remote edge (dashed), resulting in "John" having two parents. (b) includes a discontinuous unit ("gave ... up"). (c) includes a coordination construction ("John and Mary"). Pre-terminal nodes are omitted for brevity. Right: legend of edge labels.

in Figure 1a, "graduation" and "moved" are considered separate events, despite appearing in the same clause. Second, in the same example, "John" is marked as a (remote) Participant in the graduation scene, despite not being overtly marked. Third, consider the possessive construction in Figure 1c. While in UCCA "trip" evokes a scene in which "John and Mary" is a Participant, a syntactic scheme would analyze this phrase similarly to "John and Mary's children".

These examples demonstrate that a UCCA parser, and more generally semantic parsers, face an additional level of ambiguity compared to their syntactic counterparts (e.g., "after graduation" is formally very similar to "after 2pm", which does not evoke a scene). Section 6 discusses UCCA in the context of other semantic schemes, such as AMR (Banarescu et al., 2013).

Alongside recent progress in dependency parsing into projective trees, there is increasing interest in parsing into representations with more general structural properties (see Section 6). One such property is *reentrancy*, namely the sharing of semantic units between predicates. For instance, in Figure 1a, "John" is an argument of both "graduation" and "moved", yielding a DAG rather than a tree. A second property is *discontinuity*, as in Figure 1b, where "gave up" forms a discontinuous semantic unit. Discontinuities are perva-

sive, e.g., with multi-word expressions (Schneider et al., 2014). Finally, unlike most dependency schemes, UCCA uses *non-terminal nodes* to represent units comprising more than one word. The use of non-terminal nodes is motivated by constructions with no clear head, including co-ordination structures (e.g., "John and Mary" in Figure 1c), some multi-word expressions (e.g., "The Haves and the *Have Nots*"), and prepositional phrases (either the preposition or the head noun can serve as the constituent's head). To our knowledge, no existing parser supports all structural properties required for UCCA parsing.

## 3 Transition-based UCCA Parsing

We now turn to presenting TUPA. Building on previous work on parsing reentrancies, discontinuities and non-terminal nodes, we define an extended set of transitions and features that supports the conjunction of these properties.

Transition-based parsers (Nivre, 2003) scan the text from start to end, and create the parse incrementally by applying a *transition* at each step to the parser's state, defined using three data structures: a buffer $B$ of tokens and nodes to be processed, a stack $S$ of nodes currently being processed, and a graph $G = (V, E, \ell)$ of constructed nodes and edges, where $V$ is the set of *nodes*, $E$ is the set of *edges*, and $\ell : E \rightarrow L$ is the *label* function, $L$ being the set of possible labels. Some states are marked as *terminal*, meaning that $G$ is the final output. A classifier is used at each step to select the next transition based on features encoding the parser's current state. During training, an oracle creates training instances for the classifier, based on gold-standard annotations.

**Transition Set.** Given a sequence of tokens $w_1, \ldots, w_n$, we predict a UCCA graph $G$ over the sequence. Parsing starts with a single node on the stack (an artificial root node), and the input tokens in the buffer. Figure 2 shows the transition set.

In addition to the standard SHIFT and RE-DUCE operations, we follow previous work in transition-based constituency parsing (Sagae and Lavie, 2005), adding the NODE transition for creating new non-terminal nodes. For every $X \in L$, NODE$_X$ creates a new node on the buffer as a parent of the first element on the stack, with an $X$-labeled edge. LEFT-EDGE$_X$ and RIGHT-EDGE$_X$ create a new primary $X$-labeled edge between the first two elements on the stack, where the parent is

the left or the right node, respectively. As a UCCA node may only have one incoming primary edge, EDGE transitions are disallowed if the child node already has an incoming primary edge. LEFT-REMOTE$_X$ and RIGHT-REMOTE$_X$ do not have this restriction, and the created edge is additionally marked as *remote*. We distinguish between these two pairs of transitions to allow the parser to create remote edges without the possibility of producing invalid graphs. To support the prediction of multiple parents, node and edge transitions leave the stack unchanged, as in other work on transition-based dependency graph parsing (Sagae and Tsujii, 2008; Ribeyre et al., 2014; Tokgöz and Eryiğit, 2015). REDUCE pops the stack, to allow removing a node once all its edges have been created. To handle discontinuous nodes, SWAP pops the second node on the stack and adds it to the top of the buffer, as with the similarly named transition in previous work (Nivre, 2009; Maier, 2015). Finally, FINISH pops the root node and marks the state as terminal.

**Classifier.** The choice of classifier and feature representation has been shown to play an important role in transition-based parsing (Chen and Manning, 2014; Andor et al., 2016; Kiperwasser and Goldberg, 2016). To investigate the impact of the type of transition classifier in UCCA parsing, we experiment with three different models.

1. Starting with a simple and common choice (e.g., Maier and Lichte, 2016), **TUPA$_{Sparse}$** uses a linear classifier with sparse features, trained with the averaged structured perceptron algorithm (Collins and Roark, 2004) and MIN-UPDATE (Goldberg and Elhadad, 2011): each feature requires a minimum number of updates in training to be included in the model.[2]

2. Changing the model to a feedforward neural network with dense embedding features, **TUPA$_{MLP}$** ("multi-layer perceptron"), uses an architecture similar to that of Chen and Manning (2014), but with two rectified linear layers instead of one layer with cube activation. The embeddings and classifier are trained jointly.

3. Finally, **TUPA$_{BiLSTM}$** uses a bidirectional LSTM for feature representation, on top of the

---

[2]We also experimented with a linear model using dense embedding features, trained with the averaged structured perceptron algorithm. It performed worse than the sparse perceptron model and was hence discarded.

| Before Transition | | | | Transition | After Transition | | | | | Condition |
|---|---|---|---|---|---|---|---|---|---|---|
| Stack | Buffer | Nodes | Edges | | Stack | Buffer | Nodes | Edges | Terminal? | |
| $S$ | $x \mid B$ | $V$ | $E$ | SHIFT | $S \mid x$ | $B$ | $V$ | $E$ | $-$ | |
| $S \mid x$ | $B$ | $V$ | $E$ | REDUCE | $S$ | $B$ | $V$ | $E$ | $-$ | |
| $S \mid x$ | $B$ | $V$ | $E$ | NODE$_X$ | $S \mid x$ | $y \mid B$ | $V \cup \{y\}$ | $E \cup \{(y,x)_X\}$ | $-$ | $x \neq$ root |
| $S \mid y, x$ | $B$ | $V$ | $E$ | LEFT-EDGE$_X$ | $S \mid y, x$ | $B$ | $V$ | $E \cup \{(x,y)_X\}$ | $-$ | $\begin{cases} x \notin w_{1:n}, \\ y \neq \text{root}, \\ y \not\rightsquigarrow_G x \end{cases}$ |
| $S \mid x, y$ | $B$ | $V$ | $E$ | RIGHT-EDGE$_X$ | $S \mid x, y$ | $B$ | $V$ | $E \cup \{(x,y)_X\}$ | $-$ | |
| $S \mid y, x$ | $B$ | $V$ | $E$ | LEFT-REMOTE$_X$ | $S \mid y, x$ | $B$ | $V$ | $E \cup \{(x,y)_X^*\}$ | $-$ | |
| $S \mid x, y$ | $B$ | $V$ | $E$ | RIGHT-REMOTE$_X$ | $S \mid x, y$ | $B$ | $V$ | $E \cup \{(x,y)_X^*\}$ | $-$ | |
| $S \mid x, y$ | $B$ | $V$ | $E$ | SWAP | $S \mid y$ | $x \mid B$ | $V$ | $E$ | $-$ | $\text{i}(x) < \text{i}(y)$ |
| [root] | $\emptyset$ | $V$ | $E$ | FINISH | $\emptyset$ | $\emptyset$ | $V$ | $E$ | $+$ | |

Figure 2: The transition set of TUPA. We write the stack with its top to the right and the buffer with its head to the left. $(\cdot, \cdot)_X$ denotes a primary $X$-labeled edge, and $(\cdot, \cdot)_X^*$ a remote $X$-labeled edge. $\text{i}(x)$ is a running index for the created nodes. In addition to the specified conditions, the prospective child in an EDGE transition must not already have a primary parent.

dense embedding features, an architecture similar to Kiperwasser and Goldberg (2016). The BiLSTM runs on the input tokens in forward and backward directions, yielding a vector representation that is then concatenated with dense features representing the parser state (e.g., existing edge labels and previous parser actions; see below). This representation is then fed into a feedforward network similar to TUPA$_{\text{MLP}}$. The feedforward layers, BiLSTM and embeddings are all trained jointly.

For all classifiers, inference is performed greedily, i.e., without beam search. Hyperparameters are tuned on the development set (see Section 4).

**Features.** TUPA$_{\text{Sparse}}$ uses binary indicator features representing the words, POS tags, syntactic dependency labels and existing edge labels related to the top four stack elements and the next three buffer elements, in addition to their children and grandchildren in the graph. We also use bi- and trigram features based on these values (Zhang and Clark, 2009; Zhu et al., 2013), features related to discontinuous nodes (Maier, 2015, including separating punctuation and gap type), features representing existing edges and the number of parents and children, as well as the past actions taken by the parser. In addition, we use use a novel, UCCA-specific feature: number of remote children.[3]

For TUPA$_{\text{MLP}}$ and TUPA$_{\text{BiLSTM}}$, we replace all indicator features by a concatenation of the vector embeddings of all represented elements: words, POS tags, syntactic dependency labels, edge labels, punctuation, gap type and parser actions. These embeddings are initialized randomly. We additionally use external word embeddings initialized with pre-trained word2vec vectors (Mikolov

et al., 2013),[4] updated during training. In addition to dropout between NN layers, we apply word dropout (Kiperwasser and Goldberg, 2016): with a certain probability, the embedding for a word is replaced with a zero vector. We do not apply word dropout to the external word embeddings.

Finally, for all classifiers we add a novel real-valued feature to the input vector, **ratio**, corresponding to the ratio between the number of terminals to number of nodes in the graph $G$. This feature serves as a regularizer for the creation of new nodes, and should be beneficial for other transition-based constituency parsers too.

**Training.** For training the transition classifiers, we use a dynamic oracle (Goldberg and Nivre, 2012), i.e., an oracle that outputs a set of optimal transitions: when applied to the current parser state, the gold standard graph is reachable from the resulting state. For example, the oracle would predict a NODE transition if the stack has on its top a parent in the gold graph that has not been created, but would predict a RIGHT-EDGE transition if the second stack element is a parent of the first element according to the gold graph and the edge between them has not been created. The transition predicted by the classifier is deemed correct and is applied to the parser state to reach the subsequent state, if the transition is included in the set of optimal transitions. Otherwise, a random optimal transition is applied, and for the perceptron-based parser, the classifier's weights are updated according to the perceptron update rule.

POS tags and syntactic dependency labels are extracted using spaCy (Honnibal and Johnson, 2015).[5] We use the categorical cross-entropy ob-

---

[3] See Appendix A for a full list of used feature templates.

[4] https://goo.gl/6ovEhC

[5] https://spacy.io

Parser state

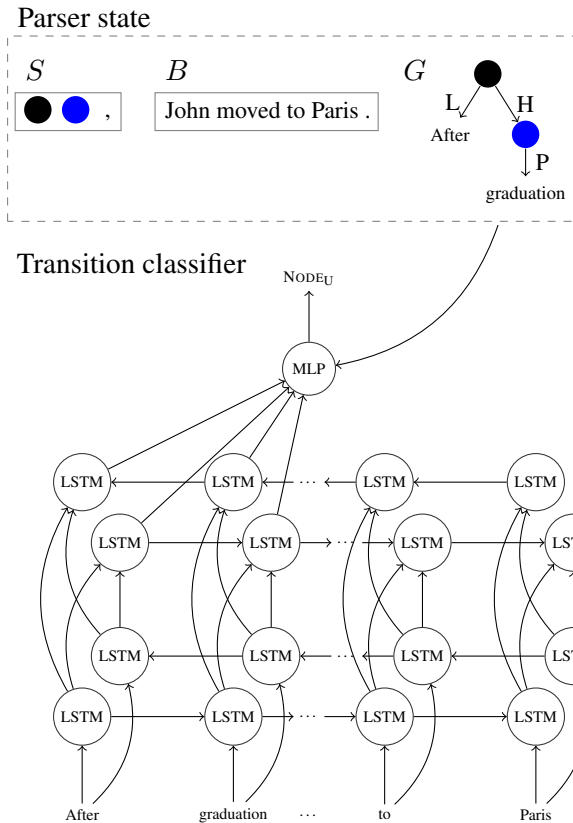

Figure 3: Illustration of the TUPA model. Top: parser state (stack, buffer and intermediate graph). Bottom: TUPA$_{\text{BiLTSM}}$ architecture. Vector representation for the input tokens is computed by two layers of bidirectional LSTMs. The vectors for specific tokens are concatenated with embedding and numeric features from the parser state (for existing edge labels, number of children, etc.), and fed into the MLP for selecting the next transition.

jective function and optimize the NN classifiers with the Adam optimizer (Kingma and Ba, 2014).

## 4 Experimental Setup

**Data.** We conduct our experiments on the UCCA Wikipedia corpus (henceforth, *Wiki*), and use the English part of the UCCA *Twenty Thousand Leagues Under the Sea* English-French parallel corpus (henceforth, *20K Leagues*) as out-of-domain data.[6] Table 1 presents some statistics for the two corpora. We use passages of indices up to 676 of the *Wiki* corpus as our training set, passages 688–808 as development set, and passages 942–1028 as in-domain test set. While UCCA edges can cross sentence boundaries, we adhere to the common practice in semantic parsing and train our parsers on individual sentences, discarding inter-relations between them (0.18% of the edges). We also discard linkage nodes and

---
[6] http://cs.huji.ac.il/~oabend/ucca.html

| | Wiki | | | 20K |
|---|---|---|---|---|
| | Train | Dev | Test | Leagues |
| # passages | 300 | 34 | 33 | 154 |
| # sentences | 4268 | 454 | 503 | 506 |
| # nodes | 298,993 | 33,704 | 35,718 | 29,315 |
| % terminal | 42.96 | 43.54 | 42.87 | 42.09 |
| % non-term. | 58.33 | 57.60 | 58.35 | 60.01 |
| % discont. | 0.54 | 0.53 | 0.44 | 0.81 |
| % reentrant | 2.38 | 1.88 | 2.15 | 2.03 |
| # edges | 287,914 | 32,460 | 34,336 | 27,749 |
| % primary | 98.25 | 98.75 | 98.74 | 97.73 |
| % remote | 1.75 | 1.25 | 1.26 | 2.27 |
| Average per non-terminal node | | | | |
| # children | 1.67 | 1.68 | 1.66 | 1.61 |

Table 1: Statistics of the *Wiki* and *20K Leagues* UCCA corpora. All counts exclude the root node, implicit nodes, and linkage nodes and edges.

edges (as they often express inter-sentence relations and are thus mostly redundant when applied at the sentence level) as well as implicit nodes.[7] In the out-of-domain experiments, we apply the same parsers (trained on the *Wiki* training set) to the *20K Leagues* corpus without parameter re-tuning.

**Implementation.** We use the DyNet package (Neubig et al., 2017) for implementing the NN classifiers. Unless otherwise noted, we use the default values provided by the package. See Appendix C for the hyperparameter values we found by tuning on the development set.

**Evaluation.** We define a simple measure for comparing UCCA structures $G_p = (V_p, E_p, \ell_p)$ and $G_g = (V_g, E_g, \ell_g)$, the predicted and gold-standard graphs, respectively, over the same sequence of terminals $W = \{w_1, \ldots, w_n\}$. For an edge $e = (u, v)$ in either graph, $u$ being the parent and $v$ the child, its yield $y(e) \subseteq W$ is the set of terminals in $W$ that are descendants of $v$. Define the set of *mutual edges* between $G_p$ and $G_g$:

$$M(G_p, G_g) =$$
$$\{(e_1, e_2) \in E_p \times E_g \mid y(e_1) = y(e_2) \wedge \ell_p(e_1) = \ell_g(e_2)\}$$

Labeled precision and recall are defined by dividing $|M(G_p, G_g)|$ by $|E_p|$ and $|E_g|$, respectively, and F-score by taking their harmonic mean. We report two variants of this measure: one where we consider only primary edges, and another for remote edges (see Section 2). Performance on remote edges is of pivotal importance in this investigation, which focuses on extending the class of graphs supported by statistical parsers.

---
[7] Appendix B further discusses linkage and implicit units.

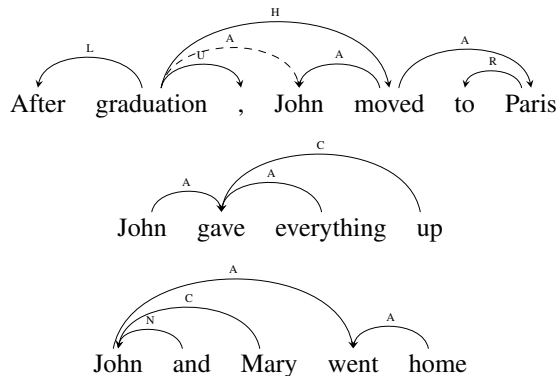

Figure 4: Bilexical graph approximation (dependency graph) for the sentences in Figure 1.

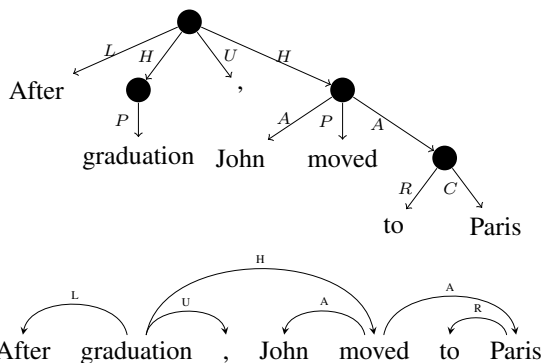

Figure 5: Tree approximation (constituency) for the sentence in Figure 1a (top), and bilexical tree approximation (dependency) for the same sentence (bottom). These are identical to the original graphs, apart from the removal of remote edges.

We note that the measure collapses to the standard PARSEVAL constituency evaluation measure if $G_p$ and $G_g$ are trees. Punctuation is excluded from the evaluation, but not from the datasets.

**Comparison to bilexical graph parsers.** As no direct comparison with existing parsers is possible, we compare TUPA to bilexical dependency graph parsers, which support reentrancy and discontinuity but not non-terminal nodes. To facilitate the comparison, we convert our training set into bilexical graphs (see examples in Figure 4), train each of the parsers, and evaluate them by applying them to the test set and then reconstructing UCCA graphs, which are compared with the gold standard.[8] In Section 5 we report the upper bounds on the achievable scores due to the error resulting from the removal of non-terminal nodes.

**Comparison to tree parsers.** For completeness, and as parsing technology is considerably more mature for tree (rather than graph) parsing, we also

perform a *tree approximation* experiment, converting UCCA to (bilexical) trees and evaluating constituency and dependency tree parsers on them (see examples in Figure 5). Our approach is similar to the tree approximation approach used for dependency graph parsing (Agić et al., 2015; Fernández-González and Martins, 2015), where dependency graphs were converted into dependency trees and then parsed by dependency tree parsers. In our setting, the conversion to trees consists simply of removing remote edges from the graph, and then to bilexical trees by applying the same procedure as for bilexical graphs.

**Baseline parsers.** We evaluate two bilexical graph semantic dependency parsers: DAGParser (Ribeyre et al., 2014), the leading transition-based parser in SemEval 2014 (Oepen et al., 2014) and TurboParser (Almeida and Martins, 2015), a graph-based parser from SemEval 2015 (Oepen et al., 2015); UPARSE (Maier and Lichte, 2016), a transition-based constituency parser supporting discontinuous constituents; and two bilexical tree parsers: MaltParser (Nivre et al., 2007),[9] and the stack LSTM-based parser of Dyer et al. (2015, henceforce "LSTM Parser"). Default settings are used in all cases. DAGParser and UPARSE use beam search by default, with a beam size of 5 and 4 respectively. The other parsers are greedy.

## 5 Results

Table 2 presents our main experimental results, as well as upper bounds for the baseline parsers, reflecting the error resulting from the conversion.

DAGParser and UPARSE are most directly comparable to TUPA$_{Sparse}$, as they also use a perceptron classifier with sparse features. TUPA$_{Sparse}$ considerably outperforms both, where DAGParser does not predict any remote edges in the out-of-domain setting. TurboParser fares worse in this comparison, despite somewhat better results on remote edges. The LSTM parser of Dyer et al. (2015) obtains the highest primary F-score among the baseline parsers, with a considerable margin.

Using a feedforward NN and embedding features, TUPA$_{MLP}$ obtains higher scores than TUPA$_{Sparse}$, but is outperformed by the LSTM parser on primary edges. However, using better input encoding, TUPA$_{BiLSTM}$ obtains substantially higher scores than TUPA$_{MLP}$ and all other parsers,

---

[8]See Appendix D for the conversion procedures.

[9]For MaltParser we use the ARCEAGER transition set and SVM classifier. Other configurations yielded lower scores.

| | Wiki (in-domain) | | | | | | 20K Leagues (out-of-domain) | | | | | |
|---|---|---|---|---|---|---|---|---|---|---|---|---|
| | Primary | | | Remote | | | Primary | | | Remote | | |
| | **LP** | **LR** | **LF** | **LP** | **LR** | **LF** | **LP** | **LR** | **LF** | **LP** | **LR** | **LF** |
| TUPA$_{\text{Sparse}}$ | 64.5 | 63.7 | 64.1 | 19.8 | 13.4 | 16 | 59.6 | 59.9 | 59.8 | 22.2 | 7.7 | 11.5 |
| TUPA$_{\text{MLP}}$ | 65.2 | 64.6 | 64.9 | 23.7 | 13.2 | 16.9 | 62.3 | 62.6 | 62.5 | 20.9 | 6.3 | 9.7 |
| TUPA$_{\text{BiLSTM}}$ | 74.4 | 72.7 | **73.5** | 47.4 | 51.6 | **49.4** | 68.7 | 68.5 | **68.6** | 38.6 | 18.8 | **25.3** |
| *Bilexical Approximation (Dependency DAG Parsers)* | | | | | | | | | | | | |
| Upper Bound | | | 91 | | | 58.3 | | | 91.3 | | | 43.4 |
| DAGParser | 61.8 | 55.8 | 58.6 | 9.5 | 0.5 | 1 | 56.4 | 50.6 | 53.4 | – | 0 | 0 |
| TurboParser | 57.7 | 46 | 51.2 | 77.8 | 1.8 | 3.7 | 50.3 | 37.7 | 43.1 | 100 | 0.4 | 0.8 |
| *Tree Approximation (Constituency Tree Parser)* | | | | | | | | | | | | |
| Upper Bound | | | 100 | | | – | | | 100 | | | – |
| UPARSE | 60.9 | 61.2 | 61.1 | – | – | – | 52.7 | 52.8 | 52.8 | – | – | – |
| *Bilexical Tree Approximation (Dependency Tree Parsers)* | | | | | | | | | | | | |
| Upper Bound | | | 91 | | | – | | | 91.3 | | | – |
| MaltParser | 62.8 | 57.7 | 60.2 | – | – | – | 57.8 | 53 | 55.3 | – | – | – |
| LSTM Parser | 73.2 | 66.9 | 69.9 | – | – | – | 66.1 | 61.1 | 63.5 | – | – | – |

Table 2: Experimental results, in percents, on the *Wiki* test set (left) and the *20K Leagues* set (right). Columns correspond to labeled precision, recall and F-score, for both primary and remote edges. F-score upper bounds are reported for the conversions. For the tree approximation experiments, only primary edges scores are reported, as they are unable to predict remote edges. TUPA$_{\text{BiLSTM}}$ obtains the highest F-scores in all metrics, surpassing the bilexical parsers, tree parsers and other classifiers.

on both primary and remote edges, both in the in-domain and out-of-domain settings. Its performance in absolute terms, of 73.5% F-score on primary edges, is encouraging in light of UCCA's inter-annotator agreement of 80–85% F-score on them (Abend and Rappoport, 2013).

The parsers resulting from tree approximation are unable to recover any remote edges, as these are removed in the conversion.[10] The bilexical DAG parsers are quite limited in this respect as well. While some of the DAG parsers' difficulty can be attributed to the conversion upper bound of 58.3%, this in itself cannot account for their poor performance on remote edges, which is an order of magnitude lower than that of TUPA$_{\text{BiLSTM}}$.

# 6 Related Work

While earlier work on anchored[11] semantic parsing has mostly concentrated on shallow semantic analysis, focusing on semantic role labeling of verbal argument structures, the focus has recently shifted to parsing of more elaborate representations that account for a wider range of phenomena.

---

[10]We also experimented with a simpler version of TUPA lacking REMOTE transitions, obtaining an increase of up to 2 labeled F-score points on primary edges, at the cost of not being able to predict remote edges.

[11]By *anchored* we mean that the semantic representation directly corresponds to the words and phrases of the text.

**Grammar-Based Parsing.** Linguistically expressive grammars such as HPSG (Pollard and Sag, 1994), CCG (Steedman, 2000) and TAG (Joshi and Schabes, 1997) provide a theory of the syntax-semantics interface, and have been used as a basis for semantic parsers by defining compositional semantics on top of them (Flickinger, 2000; Bos, 2005, among others). Depending on the grammar and the implementation, such semantic parsers can support some or all of the structural properties UCCA exhibits. Nevertheless, this line of work differs from our grammarless approach in two important ways. First, the *representations* are different. UCCA does not attempt to model the syntax-semantics interface and is thus less coupled with syntax. Second, while grammar-based *parsers* explicitly model syntax, grammarless approaches, as presented here, directly model the relation between tokens and semantic structures.

**Broad-Coverage Semantic Parsing.** Most closely related to this work is Broad-Coverage Semantic Dependency Parsing (SDP), addressed in two SemEval tasks (Oepen et al., 2014, 2015). Like UCCA parsing, SDP addresses a wide range of semantic phenomena, and supports discontinuous units and reentrancy. However, SDP uses bilexical dependencies, disallowing non-terminal nodes, useful for representing structures that have no clear head, such as co-

ordination (Ivanova et al., 2012). It also differs from UCCA in the type of distinctions it makes, which are more tightly coupled with syntactic considerations, where UCCA aims to capture purely semantic cross-linguistically applicable notions. For instance, the "poss" label in the DM target representation is used to annotate syntactic possessive constructions, regardless of whether they correspond to semantic ownership (e.g., "John's dog") or other semantic relations, such as marking an argument of a nominal predicate (e.g., "John's kick"). UCCA reflects the difference between these constructions.

Recent interest in SDP has yielded numerous works on graph parsing (Ribeyre et al., 2014; Thomson et al., 2014; Almeida and Martins, 2015; Du et al., 2015), including tree approximation (Agić and Koller, 2014; Schluter et al., 2014) and joint syntactic/semantic parsing (Henderson et al., 2013; Swayamdipta et al., 2016).

**Abstract Meaning Representation.** Another line of work addresses parsing into AMRs (Flanigan et al., 2014; Vanderwende et al., 2015; Pust et al., 2015; Artzi et al., 2015), which, like UCCA, abstract away from syntactic distinctions and represent meaning directly, using OntoNotes predicates (Weischedel et al., 2013). Events in AMR may also be evoked by non-verbal predicates, including possessive constructions.

Unlike in UCCA, the alignment between AMR concepts and the text is not explicitly marked. While sharing much of this work's motivation, not anchoring the representation in the text complicates the parsing task, as it requires the alignment to be automatically (and imprecisely) detected. Indeed, despite considerable technical effort (Flanigan et al., 2014; Pourdamghani et al., 2014; Werling et al., 2015), concept identification is only about 80%–90% accurate. Furthermore, anchoring allows breaking down sentences into semantically meaningful sub-spans, which is useful for many applications (Fernández-González and Martins, 2015; Birch et al., 2016).

Several transition-based AMR parsers have been proposed: CAMR assumes syntactically parsed input, processing dependency trees into AMR (Wang et al., 2015a,b, 2016; Goodman et al., 2016). In contrast, the parsers of Damonte et al. (2017) and Zhou et al. (2016) do not require syntactic pre-processing. Damonte et al. (2017) perform concept identification using a simple heuris-

tic selecting the most frequent graph for each token, and Zhou et al. (2016) perform concept identification and parsing jointly. UCCA parsing does not require separately aligning the input tokens to the graph. TUPA creates non-terminal units as part of the parsing process.

Furthermore, existing transition-based AMR parsers are not general DAG parsers. They are only able to predict a subset of reentrancies and discontinuities, as they may remove nodes before their parents have been predicted (Damonte et al., 2017). They are thus limited to a sub-class of AMRs in particular, and specifically cannot produce arbitrary DAG parses. TUPA's transition set, on the other hand, allows general DAG parsing.[12]

# 7 Conclusion

We present TUPA, the first parser for UCCA. Evaluated in in-domain and out-of-domain settings, we show that coupled with a NN classifier and BiLSTM feature extractor, it accurately predicts UCCA graphs from text, outperforming a variety of strong baselines by a margin.

Despite the recent diversity of semantic parsing work, the effectiveness of different approaches for structurally and semantically different schemes is not well-understood (Kuhlmann and Oepen, 2016). Our contribution to this literature is a general grammarless parser that supports multiple parents, discontinuous units and non-terminal nodes.

Future work will explore different target representations and conversion procedures (Kong et al., 2015), to compare different representations, suggesting ways for a data-driven design of semantic annotation. A parser for UCCA will enable using the framework for new tasks, in addition to existing applications such as machine translation evaluation (Birch et al., 2016). We believe UCCA's merits in providing a cross-linguistically applicable, broad-coverage annotation will support ongoing efforts to incorporate deeper semantic structures into various applications, such as sentence simplification (Narayan and Gardent, 2014) and summarization (Liu et al., 2015).

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
