# Peer review of "A Transition-Based Directed Acyclic Graph Parser for UCCA"

_ACL 2017 — decision unknown_

[Official Review · Reviewer 1 · rating 5 · confidence 4]
soundness 3 · originality 3 · clarity 5 · impact 3 · substance 4 · appropriateness 5 · meaningful comparison 3 · presentation format Oral Presentation

This paper introduces UCCA as a target representation for semantic parsing and
also describes a quite successful transition-based parser for inference into
that representation. I liked this paper a lot. I believe there is a lot of
value simply in the introduction of UCCA (not new, but I believe relatively new
to this community), which has the potential to spark new thinking about
semantic representations of text. I also think the model was well thought out.
While the model itself was fairly derivative of existing transition-based
schemes, the extensions the authors introduced to make the model applicable in
this domain were reasonable and well-explained, at what I believe to be an
appropriate level of detail.

The empirical evaluation was pretty convincing -- the results were good, as
compared to several credible baselines, and the authors demonstrated this
performance in multiple domains. My biggest complaint about this paper is the
lack of multilingual evaluation, especially given
that the formalism being experimented with is exactly one that is supposed to
be fairly universal. I'm reasonably sure multilingual UCCA corpora exist (in
fact, I think the "20k leagues" corpus used in this paper is one such), so it
would be good to see results in a language other than English.

One minor point: in section 6, the authors refer to their model as
"grammarless", which strikes me as not quite correct. It's true that the UCCA
representation isn't derived from linguistic notions of syntax, but it still
defines a way to construct a compositional abstract symbolic representation of
text, which to me, is precisely a grammar. (This is clearly a quibble, and I
don't know why it irked me enough that I feel compelled to address it, but it
did.)

Edited to add: Thanks to the authors for their response.

[Official Review · Reviewer 2 · rating 4 · confidence 5]
soundness 3 · originality 3 · clarity 5 · impact 3 · substance 4 · appropriateness 5 · meaningful comparison 3 · presentation format Oral Presentation

This paper presents the first parser to UCCA, a recently proposed meaning
representation. The parser is transition based, and uses a new transition set
designed to recover challenging discontinuous structures with reentrancies.
Experiments demonstrate that the parser works well, and that it is not easy to
build these representation on top of existing parsing approaches. 

This is a well written and interesting paper on an important problem. The
transition system is well motivated and seems to work well for the problem. The
authors also did a very thorough experimental evaluation, including both
varying the classifier for the base parser (neural, linear model, etc.) and
also comparing to the best output you could get from other existing, but less
expressive, parsing formulations. This paper sets a strong standard to UCCA
parsing, and should also be interesting to researchers working with other
expressive meaning representations or complex transition systems. 

My only open question is the extent to which this new parser subsumes all of
the other transition based parsers for AMR, SDP, etc. Could the UCCA transition
scheme be used in these cases (which heuristic alignments if necessary), and
would it just learn to not use the extra transitions for non-terminals, etc.
Would it reduce to an existing algorithm, or perhaps work better? Answering
this question isn’t crucial, the paper is very strong as is, but it would add
to the overall understanding and could point to interesting areas for future
work.

----

I read the author response and agree with everything they say.

[Official Review · Reviewer 3 · rating 4 · confidence 4]
soundness 3 · originality 3 · clarity 5 · impact 3 · substance 4 · appropriateness 5 · meaningful comparison 3 · presentation format Oral Presentation

(the authors response answer most of the clarification questions of my review)

=========================
- Summary:
=========================

The paper describes a transition-based system for UCCA graphs, featuring
non-terminal nodes,  reentrancy and discontinuities. The transition set is a
mix of already proposed transitions
(The key aspects are the swap transition to cope with discontinuities, and
transitions not popping the stack to allow multiple parents for a node.).
The best performance is obtained using as transition classifier a MLP with
features based on bidirectional LSTMs.

The authors compare the obtained performance with other state-of-the art
parsers, using conversion schemes (to bilexical graphs, and to tree
approximations): the parsers are trained on converted data, used to predict
graphs (or trees), and the predicted structures are converted ack to UCCA and
confronted with gold UCCA representations.

=========================
- Strengths:
=========================

The paper presents quite solid work, with state-of-the art transition-based
techniques, and machine learning for parsing techniques.

It is very well written, formal and experimental aspects are described in a
very precise way, and the authors demonstrate a very good knowledge of the
related work, both for parsing techniques and for shallow semantic
representations.

=========================
- Weaknesses:
=========================

Maybe the weakness of the paper is that the originality lies mainly in the
targeted representations (UCCA), not really in the proposed parser.

=========================
- More detailed comments and clarification questions:
=========================

Introduction

Lines 46-49: Note that "discontinuous nodes" could be linked to
non-projectivity in the dependency framework. So maybe rather state that the
difference is with phrase-structure syntax not dependency syntax.

Section 2:

In the UCCA scheme's description, the alternative "a node (or unit) corresponds
to a terminal or to several sub-units" is not very clear. Do you mean something
else than a node is either a terminal or a non terminal? Can't a non terminal
node have one child only (and thus neither be a terminal nor have several
sub-units) ?

Note that "movement, action or state" is not totally appropriate, since there
are processes which are neither movements nor actions (e.g. agentless
transformations).
(the UCCA guidelines use these three terms, but later state the state/process
dichotomy, with processes being an "action, movement or some other relation
that evolves in time").

lines 178-181: Note that the contrast between "John and Mary's trip" and "John
and Mary's children" is not very felicitous. The relational noun "children"
evokes an underlying relation between two participants (the children and
John+Mary), that has to be accounted for in UCCA too.

Section 4:

Concerning the conversion procedures:
- While it is very complete to provide the precise description of the
conversion procedure in the supplementary material, it would ease reading to
describe it informally in the paper (as a variant of the
constituent-to-dependency conversion procedure à la Manning, 95). Also, it
would be interesting to motivate the priority order used to define the head of
an edge.

- How l(u) is chosen in case of several children with same label should be made
explicit (leftmost ?).

- In the converted graphs in figure 4, some edges seem inverted (e.g. the
direction between "John" and "moved" and between "John" and "gave" should be
the same).

- Further, I am confused as to why the upper bound for remote edges in
bilexical approximations is so low. The current description of the conversions
do not allow to get an quick idea of which kind of remote edges cannot be
handled.

Concerning the comparison to other parsers:
It does not seem completely fair to tune the proposed parser, but to use
default settings for the other parsers.

Section 5

Line 595: please better motivate the claim "using better input encoding"

Section 6

I am not convinced by the alledged superiority of representations with
non-terminal nodes. Although it can be considered more elegant not to choose a
head for some constructions, it can be noted that formally co-head labels can
be used in bilexical dependencies to recover the same information.